# Cannabinoids in the Modulation of Oxidative Signaling

**DOI:** 10.3390/ijms24032513

**Published:** 2023-01-28

**Authors:** Cristina Pagano, Beatrice Savarese, Laura Coppola, Giovanna Navarra, Giorgio Avilia, Chiara Laezza, Maurizio Bifulco

**Affiliations:** 1Department of Molecular Medicine and Medical Biotechnology, University of Naples “Federico II”, Via Pansini 5, 80131 Naples, Italy; 2Institute of Endocrinology and Experimental Oncology (IEOS), National Research Council (CNR), Via Pansini 5, 80131 Naples, Italy

**Keywords:** cannabinoids, ROS, oxidative stress, neurodegeneration, cancer, inflammation

## Abstract

Cannabis sativa-derived compounds, such as delta-9-tetrahydrocannabinol (THC) and cannabidiol (CBD), and components of the endocannabinoids system, such as N-arachidonoylethanolamide (anandamide, AEA) and 2-arachidonoylglycerol (2-AG), are extensively studied to investigate their numerous biological effects, including powerful antioxidant effects. Indeed, a series of recent studies have indicated that many disorders are characterized by alterations in the intracellular antioxidant system, which lead to biological macromolecule damage. These pathological conditions are characterized by an unbalanced, and most often increased, reactive oxygen species (ROS) production. For this study, it was of interest to investigate and recapitulate the antioxidant properties of these natural compounds, for the most part CBD and THC, on the production of ROS and the modulation of the intracellular redox state, with an emphasis on their use in various pathological conditions in which the reduction of ROS can be clinically useful, such as neurodegenerative disorders, inflammatory conditions, autoimmunity, and cancers. The further development of ROS-based fundamental research focused on cannabis sativa-derived compounds could be beneficial for future clinical applications.

## 1. Introduction

As has been broadly reported in the literature and in books, Cannabis sativa and its compounds—called cannabinoids or phytocannabinoids—have been widely used for both recreational and medicinal purposes [1].

The practice of using cannabis as a medical preparation has ancient roots: as early as 2600 BC, Chinese emperors were known to be utilizing cannabis plants in order to relieve cramps, and rheumatic and menstrual pains. Despite its numerous beneficial effects, cannabis for recreational use is still prohibited in many countries, and marijuana is still considered an illicit drug in many parts of the world [2].

Over the years, more than 100 compounds known as phytocannabinoids have been described: the most abundant are delta-9-tetrahydrocannabinol (THC) and cannabidiol (CBD). In 1964, THC’s structure was identified, and this compound was recognized as the major psychoactive component of cannabis [2]. These natural compounds induce a number of effects including euphoria, analgesia, and appetite stimulation, alongside their antiemetic and anti-inflammatory effects [3]. The identification of THC and CBD led to the discovery of cannabinoid receptors, and this contributed to the recognition of the existence of the endocannabinoid system (ECS) in mammalian organisms. In addition to phytocannabinoids and endocannabinoids (ECs), other promising compounds are the synthetic cannabinoids (SCBs) produced by chemical synthesis. These compounds exert their function by binding with great selectivity and potency to cannabinoid receptors and reproducing effects comparable to those induced by phytocannabinoids and ECs [3]. The main goal of this review is to investigate the potential antioxidative effects of cannabinoid compounds in several diseases and conditions, such as neurodegenerative disorders, inflammation, pathologies of the immune system, and cancer, and to consider the increase in reactive oxygen species (ROS) reported in these pathological conditions [4,5]. We therefore analyzed the antioxidant properties of cannabinoids as modulators of intracellular ROS production and redox state (Figure 1).

### 1.1. The Endocannabinoid System

In the 1990s, one of the major topics investigated in this field was the two cannabinoid receptors (CB1 and CB2) and their endogenous ligands, known as endocannabinoids [3]. As a whole, this complicated system is known as the endocannabinoid system or the endocannabinoidome [6]. Both receptors are G protein-coupled receptors (GPCRs), and their activation inhibits the activity of adenylyl cyclase; this affects the cytoplasmatic concentration of cyclic adenosine monophosphate (cAMP) and the consequent stimulation of multiple signaling pathways, including mitogen-activated protein kinase (MAPK), phosphoinositide 3-kinase (PI3K), and cyclooxygenase (COX) 2 pathways [3]. The distribution of CB1 and CB2 receptors is well described and accounts for the recognized psychotropic and peripheral effects of THC: CB1 is highly abundant in the central nervous system (CNS) and in the peripheral nervous system (PNS), while the presence of CB2 receptors is predominantly restricted to immune tissues and cells [2]. ECs such as N-arachidonoylethanolamide (anandamide, AEA) and 2-arachidonoylglycerol (2-AG) consist of long-chain polyunsaturated fatty acids, amides, esters, and ethers, inducing biological effects following their interaction with CB receptors and non-CB receptors alike. ECs are synthesized on demand originating from membrane lipids, owing to the action of enzymes of the lipase class. In particular, AEA is produced from the hydrolysis of N-arachidonoyl phosphatidyl ethanolamine (NAPE) by a NAPE-PLD enzyme. The generation of 2-AG mainly involves two enzymes, diacylglycerol lipase-α and -β (DAGLα and DAGLβ), through the hydrolysis of arachidonoyl-containing diacylglycerols (DAGs). Immediately after synthesis, both anandamide and 2-AG are released in the extracellular space where they bind to CBs present on neighboring cells. Once their biological action is complete, ECs are inactivated by a regulated degradation process, which is principally hydrolysis. The enzyme responsible for the hydrolysis of AEA is fatty acid amide hydrolase (FAAH) while 2-AG is degraded to glycerol and arachidonic acid by monoacyl glycerol lipase (MAGL) [7].

There have been numerous studies assessing the action of ECs as inflammation and metabolism modulators, as well as neuromodulators. Dysregulation of the endocannabinoid system has been associated with several diseases, such as neurodegenerative disorders, multiple sclerosis, inflammation, and cancer [8]. Prior research suggests that AEA has many potential therapeutic effects in the treatment of cancer: not only has it been reported to inhibit the proliferation of the breast cancer cell line MCF-7 through its binding to CB1 receptors [9], but it has also been reported to inhibit many other tumor types such as colorectal, thyroid, skin, and pancreatic cancers [10,11].

The pharmacological actions of cannabinoids are not exclusively limited to CB1 and CB2, but many authors have reported their ability to target an extensive range of receptors: among these, we report the peroxisome proliferator-activated receptors (PPARs), the transient receptor potential cation channel subfamily V member 1 (TRPV1), the G-protein-coupled receptor 55 (GPR55), and the 5-hydroxytryptamine receptor subtype 1A (5-HT1A) [12].

### 1.2. Oxidative Stress and Reactive Oxygen Species

The preservation of redox homeostasis, cell cycle signaling, and hormone production rely on the maintenance of controlled levels of intracellular ROS [13]. ROS is a group of highly reactive oxygen radicals and molecules, characterized by strong oxidizing properties and generally removed by the antioxidant mechanisms of the cell. The most abundant ROS are the superoxide anion (O2 •−), hydrogen peroxide (H_2_O_2_), and the hydroxyl radical (HO•) [14]. Oxidative stress is characterized by the impaired production of reactive oxygen/nitrogen species (ROS/RNS); this has harmful effects on the body and plays a key role in affecting a number of physiological functions. These highly reactive compounds interact with biomolecules, especially membrane lipids, which leads to different biochemical reactions such as peroxidation of the polyunsaturated fatty acids, nitration and carbonylation of proteins, and oxidation of DNA. All of these converge in cell death [15]. There have been numerous studies investigating the reduced activity of the antioxidant system caused by compromised enzymatic components such as superoxide dismutase (SOD), catalase (CAT), and glutathione peroxidase (GPx) [16]. A large production of reactive ROS and oxidative stress have a pathological role in inflammatory diseases such as multiple sclerosis (MS), inflammatory bowel disease (IB), rheumatoid arthritis (RA), and atherosclerosis. In addition, during infection, blood immune cells are stimulated in order to produce ROS via the NADPH oxidase 2 (NOX2) complex as a defense mechanism against pathogens [14]. Chronic granulomatous diseases (CGD) are usually the result of altered ROS production in defective phagocyte events, resulting in severe and recurrent infections.

Nowadays, it is widely reported that cannabinoids are able to influence and modulate mitochondrial functions and dynamics, altering different biological processes such as the regulation of intracellular calcium levels, bioenergetic metabolism, apoptosis, mitochondrial respiratory electron transport chain activity, and the production of mitochondrial reactive oxygen species [17].

Many authors agree in regarding CBD as an appealing therapeutic agent for neuroimmune disorders, as it can affect redox state in many ways [14] (Figure 2).

### 1.3. Methodology

Our manuscript summarizes the recent extensive literature describing the effects of cannabinoids on ROS production related to various pathologies where their modulation might be of clinical importance. The research of the literature was conducted using search terms and obtaining query results from different databases including PubMed, Google Scholar and Web of Science. In particular, we selected all relevant studies published between 2010 and 2022, and approximately 150 articles were retrieved from databases. Finally, 87 articles were considered and deemed relevant for this review. Data gathering was performed using terms such as ‘cannabinoids and ROS production’, ‘cannabinoids/ECs and disease’, ‘neurodegenerative disease and CBD’, ‘ROS and cancer’, ‘CBD and immune system’, ‘CBD and inflammation’, and ‘ROS and inflammation’. Among those included in the text, we have reported in Table 1 some that summarize the effects of cannabinoids through the modulation of ROS.

## 2. Cannabinoids and ROS Modulation in Neurodegenerative Diseases

The majority of neurological disorders are characterized by biochemical and structural abnormalities and alterations that affect both the central nervous system and peripheral nervous system. Diseases of this type present a wide range of clinical symptoms: among these, partial or complete paralysis, muscle weakness, loss of sensation, seizures, and poor cognitive abilities are reported. Currently, some of the most common and frequently studied neurological disorders are Alzheimer’s disease (AD), Huntington’s disease (HD), epilepsy, MS, neuropathic pain, and Parkinson’s disease (PD), and in parallel, over time, an extensive literature has developed recording how cannabis and cannabinoids could provide benefits in many neurological disorders in humans [4]. In fact, the effects of ECs and phytocannabinoids on the central nervous system have been discussed by many authors in the literature: recently, cannabinoid receptors and their agonists have been shown to play a modulatory role on the reduction of the activation of the N-methyl-D-aspartate receptor (NMDAr) [28].

Nowadays, it is widely reported that, in many neurological disorders, a hyper-activation of microglia cells is present, leading to the increased secretion of neurotoxic agents such as ROS. In recent decades, a number of authors have recognized in CBD a fundamental and promising neuroprotective effect in neurological disorders: this is made possible by the targeting of microglia-mediated neuroinflammation, reducing some proinflammatory cytokines, reactive oxygen species, and the levels of other neurotoxic factors [29]. In 2019, Dos-Santos-Pereira et al. conducted experiments on the effects of CBD on induced microglia-mediated neuroinflammation after the administration of lipopolysaccharide (LPS), with the consequent activation of toll-like receptor 4 (TLR4), resulting in reactive oxygen species production attributable to the activation of the enzyme nicotinamide adenine dinucleotide phosphate (NADPH) oxidase [18]. The increase in ROS leads to the activation and transduction of NFkB, with the subsequent production of pro-inflammatory cytokines, such as TNF-α, IL-6, IFN-β and IL-1β. As reported, CBD showed antioxidant and anti-inflammatory properties, with the potential to reduce ROS production, proinflammatory cytokines secretion, and NFkB signaling [18]. It is well known that high ROS concentrations lead to the damage of macromolecules, especially to membrane lipids, in a phenomenon known as lipid peroxidation. This is particularly important because the chemical integrity of the central nervous system is the main factor in ensuring the proper functioning of the brain. Bearing in mind the high oxygen levels in the brain and the abundant polyunsaturated fatty acids in neural membrane, the inclination of brain cells to accumulate ROS is readily understandable. Neurodegenerative disease could be caused by a biochemical alteration due to oxidative stress [30]. Recent findings have suggested a potential role for CBD in the prevention of Alzheimer’s disease. AD is a widespread neurological disorder characterized by the accumulation of Aβ plaques. Recent studies reported that the agglomeration of Aβ plaques is directly linked with microglial activation and the consequential secretion of cytokine. In fact, increased ROS and lipid peroxidation are also reported, and cannabidiol neuroprotective and anti-oxidative properties that are effective against amyloid peptide toxicity have been preliminarily demonstrated [31,32]. Moreover, preliminary experiments on Caenorhabditis elegans (*C. elegans*) demonstrated that CBD could be of benefit in treating AD [33]. As reported by Harvey et al., anandamide and cannabidiol have the ability to protect against oxidative damage in neuronal cell lines, but their influence on the toxic hallmark of Alzheimer’s, Aβ protein plaques, is still unknown [34].

Moreover, the multi-targeted efficacy of the ECs also emerges as a potential approach for the treatment of AD in hyperglycemic conditions. ECs and synthetic cannabinoids can exert protective actions influencing a number of physiological responses, such as neuronal plasticity, neuroprotection, redox homeostasis against oxidative stress, and mitochondrial dysfunction induced in hippocampal neurons by hyperglycemia [35].

Parkinson’s disease is the second most common neurodegenerative disease after AD, and it is associated with a progressive loss of dopaminergic neurons in the substantia nigra of the midbrain, resulting from the intracellular accumulation of Lewy bodies containing the misfolded protein α-synuclein. Although current treatments using L-DOPA and deep brain stimulation may relieve symptoms, there is currently no therapeutic means of halting the progression of neurodegeneration. To this end, mitochondrial dysfunction and oxidative stress are assumed to play a key role in the pathogenesis of PD, and recent studies indicate that antioxidant phytocannabinoids such as CBD may exhibit potential therapeutic benefits due to their neuroprotective effect, in addition to improving the antioxidant defense of striatal neurons projected towards the substantia nigra [36]. Huntington’s disease is a neurodegenerative disease caused by the accumulation of mutant huntingtin protein in neurons and glial cells. Downregulation of the CB1 receptor has been detected in early HD patients. CB1 could inhibit the release of glutamate, and downregulation of the CB1 receptor would lead to a high level of glutamate and aggravate its damage to nerve cells [36]. Multiple studies have shown that the downregulation of CB1 may also aggravate cytotoxicity such as oxidative stress. Experiments conducted in HD model mice reported the neuroprotective effect of cannabigerol (CBG). Additionally, after administration of selected CB2 receptor agonists, inflammation, brain edema, movement disorders, and neuronal loss were reduced [19]. As evidenced by previous studies, cannabinoids can bind to CB1/2 receptors to alleviate neurotoxicity, oxidative damage, and inflammation, with THC (10 mg/kg·bw) and CBD (10 mg/kg·bw) considered a new therapeutic avenue for HD. However, clinical evaluation is still needed before cannabinoids can be used as a treatment for HD, where Sativex^®^ (THC + CBD) could be a promising choice [19]. Moreover, increased ROS concentration has been shown to promote increased permeability of the blood brain barrier (BBB), resulting in the derangement of iron metabolism in the brain and the downregulation of cannabinoid type 2 receptors. In summary, the loss of integrity results in the activation of signaling pathways leading to another proinflammatory change of the microglial phenotype. This change contributes to increased neuronal apoptotic activity, leading to cognitive deficits that are characteristic of neurodegenerative diseases [37]. Owing to the low toxicity of CBD in humans, interesting studies have been conducted to determine its clinical efficacy in various pathologies. To date, there have been approximately 2500 clinical trials focused on demyelinating disorders, and among them only 30 studies have been related to the benefits of cannabis or cannabinoids. There are 19 clinical studies addressing the effect of CBD in demyelinating disorders, particularly MS. Patients with MS have disturbances in homeostasis reduction-oxidation (redox), such as alterations in the activity of oxidative and antioxidant enzymes and the presence of products of degradation. Navarrete et al. reported that medical cannabis has been studied as a potential therapy for MS symptomatology, including pain and spasticity [38]. However, the improvements noted in mobility and pain management suggested that cannabinoids, as promising antioxidants, could be clinically useful for the treatment of those neurodegenerative diseases that still remain incurable today (Figure 3).

## 3. Cannabinoids and ROS Modulation in Cancer

Several studies suggest that CBD and its derivatives can have diverse pharmacological effects on many different diseases and conditions, including cancer [39]. This disease is usually characterized by uncontrolled proliferation due to DNA mutations and apoptosis inhibition [40]. Nowadays, cannabinoids are principally used as palliative therapy, aiming to relieve the side effects of chemotherapy, such as muscle spasms, intractable nausea, and cachexia [11], but numerous studies of cell cultures and animals reveal anti-tumor effects of cannabinoids in various cancer types [41].

Several studies suggest that CBD, in addition to providing other inhibitory effects on tumor migration, angiogenesis, metastasis, and inflammation, also has the potential to inhibit cell proliferation while enhancing tumor death [42]. Among a variety of phytocannabinoids, two are the most promising for a therapeutic approach: THC and cannabidiol CBD. Equally, previous studies have also emphasized how the major endocannabinoids, AEA and 2-AG, have the potential to induce death in tumoral cells [3]. In the last decade, a number of authors have recognized the potential beneficial effects of phytocannabinoids and endocannabinoids on cancer cell proliferation. For instance, THC and CBD may decelerate tumor progression in breast cancer patients, inducing an arrest in cell cycle progression and cell growth and enhancing cancer cell apoptosis by inhibiting constitutive active pro-oncogenic signaling pathways [43]. Correspondingly, it has been reported that CBD has the ability to suppress the activation of the EGF/EGFR signaling transduction pathway and all its downstream targets (such as Akt, ERK, and NF-kB). This supports the hypothesis that the Cannabis sativa extract might play a pivotal role in the modulation of growth factors, inflammation, and cell growth [12].

In 2018, Fonseca et al. studied the effects of cannabinoids and endocannabinoids on endometrial cancer cell viability by analyzing the expression levels of cannabinoid receptors, TRPV1, and endocannabinoid-metabolizing enzymes [44]. Previous studies have shown how THC could decrease intracellular ROS production and activate glutathione (GSH) simultaneously in oral cancer, inducing tumor cell hyper-proliferation. Through several different molecular mechanisms such as apoptosis, autophagy, and the reduction of oxidative stress, the cannabinoids were able to decrease the growth of oral cancer cells by blocking the cell cycle progression [45]. It is now common knowledge that the production of ROS primarily occurs in mitochondrial electron transport. This impairment in ROS production stimulates dysfunction in the organelle by damaging its membrane and potentially leads to electron leakage and consequent apoptosis. Jeong et al., in 2019, demonstrated that CBD induces mitochondrial ROS overproduction in colorectal cancer (CRC), acting on mitochondrial complex I and IV, even if the molecular mechanism underlying the increased production after CBD treatment is still unveiled [20]. It is well known that the Warburg effect and mitochondrial dysfunction are reported in many different cancer types [46,47], and this is usually highly correlated with prolonged ROS accumulation in cancer cells. The role of the intracellular antioxidant system is to counteract a certain level of ROS with antioxidants, aiming to balance the intracellular redox homeostasis [48]. To date, the therapeutic potential of cannabis has been examined for the most part in preclinical studies on a wide range of cancers, and the clinical studies are limited to very few tumor types. To our knowledge, few clinical trials have been proposed on recurrent glioma that is resistant to chemotherapy and radiotherapy [23].

## 4. Cannabinoids and ROS Modulation in Inflammatory Diseases

Oxidative stress brought on by excessive ROS production is a crucial part of the immune system’s defense against pathogens and stimulates tissue restoration. However, metabolic changes brought on by excessive ROS production also produce many adverse effects and can cause the onset or worsening of a number of illnesses. It is thought that the therapeutic control of oxidative stress in numerous illnesses may be caused by the endocannabinoid system. A variety of preclinical models have recently shown multidirectional biological effects, including cannabidiol’s antioxidant and anti-inflammatory properties [49]. It is also well known that ROS play a key role at each step of atherogenic plaque formation [21]. It is important to highlight the fact that all of these start with the accumulation of low-density lipoproteins (LDL) in the intima of the arterial vessel, where the production of RNS and ROS enhance LDL oxidation: this is the oxidative theory of atherosclerosis [50]. From these results, it is clear how complex the role of ROS is in regulating inflammation in disease. CBD can affect the redox balance, acting as an antioxidant, interfering with the production of free radicals, and making them less dangerous. Not only has it been reported that CBD impairs the activity of two important enzymes involved in the formation of superoxide radicals, the xanthine oxidase (XO) and NADPH oxidase (NOX1 and NOX4) [48], but also that it is capable of the chelation of metals ions required in the Fenton reaction during the generation of ROS [51]. Cannabinoids exert their potential role in the immune system via interactions with both CB1 and CB2 receptors, modulating the inflammation underlying atherogenesis [52]. In lean hyperlipidemic E3L.CETP mice, rimonabant prevents the development of atherosclerosis, which is explained by a decrease in hepatic very low density lipoprotein (VLDL) particle formation and an increase in VLDL turnover that lowers plasma non-HDL-C. CB1 receptor antagonism may be a potential method of preventing cardiovascular diseases, possibly through decreasing lipids as well as inflammation, since E3L.CETP mice are a well-established model for human-like lipoprotein metabolism and atherosclerosis formation [53]. Elsewhere, cannabinoids could be considered a potential remedy in metabolic syndrome [54]. ECS has effects on numerous organs; recent researches have intensified their focus on the potential role of the ECS in kidney diseases [55]. MRI 1867 is an orally bioavailable dual-targeted drug that can pharmacologically treat obesity-related chronic kidney disease by inhibiting both the CB1 receptor and the inducible nitric oxide synthase (iNOS) signaling pathways [56]. Recently, it has been observed that cisplatin-induced nephropathy is restricted by the endocannabinoid system through CB2 receptors, thus reducing inflammation, oxidative/nitrosative stress, and cell death. Selective CB2 agonists may therefore represent a novel and promising strategy for preventing this life-threatening side effect of chemotherapy [57]. Inflammatory bowel disease produces abnormally high amounts of ROS, which may play a significant role in causing tissue harm. Treatment for IB disease may benefit from the use of antioxidants with added anti-inflammatory qualities. In both baseline and oxidative stress settings, CBD lowers oxidative stress by preventing the overproduction of ROS species that are harmful to epithelial cells [22]. In an IB disease murine experimental model, the non-psychotropic plant cannabinoid CBG had protective benefits. The action of CBG was linked to altered cytokine levels (IL-1b, IL-10, and interferon-g) and decreased expression of iNOS (but not COX-2). Studies on peritoneal macrophages indicate that CBG reduces the synthesis of nitric oxide that is produced from the iNOS, and that the cannabinoid CB2 receptors may have a detrimental influence on this result. Additionally, CBG demonstrates antioxidant effects on intestinal epithelial cells that have experienced oxidative stress as well as on inflammatory gut tissue [58]. Additionally, it has been observed that the agonistic activity of the CB2 receptor in the healing of intestinal ischemia and reperfusion injury is associated with anti-inflammatory mechanisms that include the inhibition of inflammatory polymorphonuclear cell migration, which is the source of acute and initial responses of inflammation, the inhibition of production of provocative and pro-inflammatory cytokines such as TNF-a and IL-1b, and the rebalancing of the oxidant/antioxidant redox system [59]. There is strong evidence that oxidative stress plays a role in the etiology of the chronic skin condition psoriasis [60,61]. Cannabinoids are able to inhibit the proliferation of keratinocytes, angiogenesis, and inflammation in psoriasis pathogenesis [62]. Furthermore, the ECS has numerous interactions with the innate immune system. Because of the pro-oxidative circumstances and unique receptor profiles associated with psoriasis, psoriatic neutrophils are more prone to NETosis than neutrophils from healthy individuals. As a result, CBD consumption could have a beneficial impact and lower NETosis in psoriatic neutrophils [63]. An article published in 2018 showed that granulocytes taken from psoriasis vulgaris patients have greater levels of Nrf2 pathway activation and CB2 expression, which may reduce oxidative stress and operate as an anti-inflammatory strategy, but which may also point to increased oxidative phospholipid changes [64]. By suppressing inflammation-related cytokines including IL-6 and TNF-α, CB2 receptor activation by the agonist AM1241 ameliorated Bleomycin-induced lung fibrosis and reduced fibrosis by lowering levels of hydroxyproline and collagen type 1 [65].

## 5. Cannabinoids and ROS Modulation in the Immune System

The ECS has numerous connections with the cells of the immune system (IS) and has been shown to be able to regulate its various functions.

As previously noted, the main receptor involved in the immuno-modulatory effects of ECS is CB2, whose expression is predominant on immune tissues, such as lymph nodes and spleen, and on immune cells, primarily on B-cells and NK cells [2]. TRPV1 channels have also been found on cells of the immune system, namely B and T-lymphocytes, NK cells, and >99% of primary CD3 T-cells, where it regulates their activation and proliferation rate through the transport of [Ca^2+^]. Macrophages have also been found to express this type of channel, although in this case, TRPV1 activation has been associated with an inhibitory effect [66].

As such, the ECS plays a key role in the proper functioning of the immune system on several levels. It has been established in studies that its secretion function is essential for the migration and retention of hematopoietic stem and progenitor cells, for the trafficking and mobilization of mature immune cells, and for the functions of effector cells, and also for the general regulation of both innate and adaptive immunities [67].

One of the most peculiar functions of the IS is its ability to generate ROS to neutralize pathogens [3]. Consequently, ROS production is vital for innate immunity, where cells with a phagocytic function such as macrophages and neutrophils can activate a “respiratory burst” for their antimicrobial activity against bacteria and parasites [68].

It has been shown that CB1 is able to regulate the production of pro-inflammatory ROS, which, through the activation of the Ras family small G protein Rap1, is negatively regulated by CB2 [69]. Even the production of ROS by neutrophils could be affected by ECS: through its activation, GPR55 is capable of inhibiting its production as well as inhibiting neutrophil degranulation [70]. Furthermore, it has been shown that ECS can influence the immune homeostasis in the gut by promoting the presence of CX3CR1hi macrophages, which are immunosuppressive. Moreover, it is also able to exert neuroprotective and anti-inflammatory effects on the brain [24,25].

For these reasons, and because of their ability to regulate ROS, cannabinoids have become particularly interesting as a subject for study with regard to autoimmune diseases (ADs), which are a group of pathologies arising from an abnormal immune response against self-antigens. The persistent auto-activation of immune cells leads to an increased production of ROS, which represents one of the main reasons for the extensive damage to healthy tissue typical of these conditions.

In an article published in 2021, an analysis was carried out via MeSH on 677 journal articles; data from experimental AD animal models revealed that both natural and synthetic cannabis suppress inflammatory responses mediated by immune cells that are responsible for the recurrence and progression of AD [26].

Type 1 diabetes mellitus (T1DM) is a major subtype of diabetes characterized by is insulin deficiency due to the autoimmune destruction of insulin-producing beta cells in the pancreas.

Several studies have been published on the possible ability of cannabinoids to improve oxidative stress in diabetes, and thus also to improve cardiovascular health. In 2022, a study carried out on mice was published showing that, compared to diabetic WT mice, diabetic CB2R−/− animals had worse oxidative stress, inflammation, apoptosis, and fibrosis, indicating a critical protective role for CB2R in mitigating diabetes-induced cardiac tissue injury. Selective CB2R activation reduces myocardial inflammation, tissue damage, and fibrosis brought on by diabetes, while maintaining the heart’s functional contractile ability [71]. A 2018 study that examined the effects of cannabinoids on diabetic rats found that it increased myocardial GPR18 expression and, at least in part, prevented LV dysfunction by reversing diabetes-induced increases in cardiac vagal dominance and myocardial oxidative stress, as well as decreases in the levels of circulating NO and in the phosphorylation of myocardial eNOS, Akt, and ERK1/2 [72].

THC has also been shown to improve the cardiovascular health of STZ-Diabetic Wistar-Kyoto rats by reducing oxidative stress, lipid peroxidation, blood glucose level lipid peroxidation, and blood glucose levels [73]. The beneficial effects of CBD on autoimmune diseases and pathological memory T-cells were also shown by means of microarray-based gene expression profiling, which also explains how CBD exerts its anti-inflammatory effects [27].

A further example could be RA, which is a chronic synovial illness that arises from a systemic immune dysregulation characterized by an infiltrate of immune cells, primarily leukocytes, in joints. Here, their altered activation can lead to a powerful burst of oxidation due to the presence of pro-inflammatory factors. The over-generation of ROS and the resulting oxidative stress may be one of the leading causes of RA, as demonstrated by an overabundance of ROS in synovial tissues and fluids [74]. Consequently, the targeting of ROS could be beneficial. It has been recorded that both CB1 and CB2 are upregulated in the synovial membrane of RA-affected patients. Consequently, a cannabinoid-based therapy can be beneficial and can be applied both by modulating the activity of receptors, and by inhibiting the enzymes responsible for the degradation of endocannabinoids, such as FAAH. Indeed, the use of CB1 antagonists and FAAH-inhibitors in vivo has already been shown to result in anti-arthritic effects [75,76,77]. At the same time, CB2 receptor activation and TRPV1 antagonism decrease synovitis and RA symptoms by reducing inflammation [78,79,80]. The modulation of the ECS would thus lead to an inactivation of the immune cells responsible for the production of the ROS and for the general damage to the tissues.

Similarly, both multiple sclerosis, a chronic autoimmune demyelinating disease, and inflammatory bowel disease, which includes various gastrointestinal inflammatory pathologies, could benefit from the anti-inflammatory and immunomodulatory effects of the ECS. In MS, it has been found in studies that the activated microglia and infiltrated macrophages produce vast amounts of ROS and oxidative injury; so much so, that oxidative stress has been shown to be involved in several MS pathological hallmarks [81]. Elliot et al. and Nichols et al. both conducted studies on the anti-inflammatory and neuroprotective effects of CBD on in vivo models of autoimmune encephalomyelitis (EAE), concluding that the administration of the drug was beneficial for the overall neuro-inflammation that is at the basis of ROS production [82,83].

In IBD, oxidative stress generated by chronically activated phagocytic immune cells may be implicated in the pathogenesis of the disease as much as in the neoplastic transformation that the affected tissues may undergo [84]. The ECS plays a generally protective role in the gastrointestinal tract, controlling its motility and secretion activity. Indeed, activation of CB1 and CB2 reduces colonic inflammation [85]; similarly, the inhibition of FAAH and of the endocannabinoid membrane transport (VDM11) confers protection against gastrointestinal tract inflammation by raising the ECS levels [86].

## 6. Conclusions

This analysis leads to the conclusion that ROS play a pivotal role in neuroinflammation, peripheral immune responses, and pathological processes such as cancer. This analysis also reviews the way in which CBD readily targets oxidative signaling and ROS production. The overproduction of ROS that generates oxidative stress plays a physiological role in mammalian cells, but a disequilibrium can lead to negative outcomes, such as the development and/or the exacerbation of many diseases. Future studies could fruitfully explore the involvement of G-protein coupled receptors and their endogenous lipid ligands forming the endocannabinoid system as a therapeutic modulator of oxidative stress in various diseases. A further interesting research topic is the contribution of phytocannabinoids in the modulation of oxidative stress. In future work, investigating the biochemical pathways in which CBD functions might prove important. As reported before, CBD exhibited a fundamental and promising neuroprotective role in neurological disorders, reducing proinflammatory cytokine production in microglia and influencing BBB integrity. Previous studies have also emphasized the antiproliferative role of CBD on cancer cells and its impairment of mitochondrial ROS production. In conclusion, it has been reported that cannabinoids modulate oxidative stress in inflammation and autoimmunity, which makes them a potential therapeutic approach for different kinds of pathologies.

## Figures and Tables

**Figure 1 ijms-24-02513-f001:**
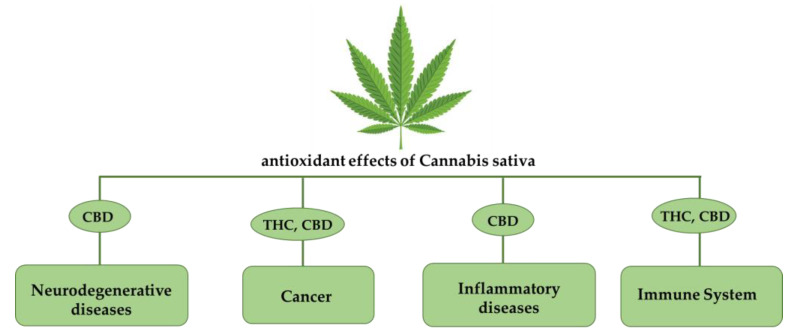
**Schematic representation of the antioxidant effects of Cannabis sativa derivatives.** Both of the two main phytocannabinoids, THC and CBD, have been found to be beneficial to different classes of pathologies owing to their antioxidant effects.

**Figure 2 ijms-24-02513-f002:**
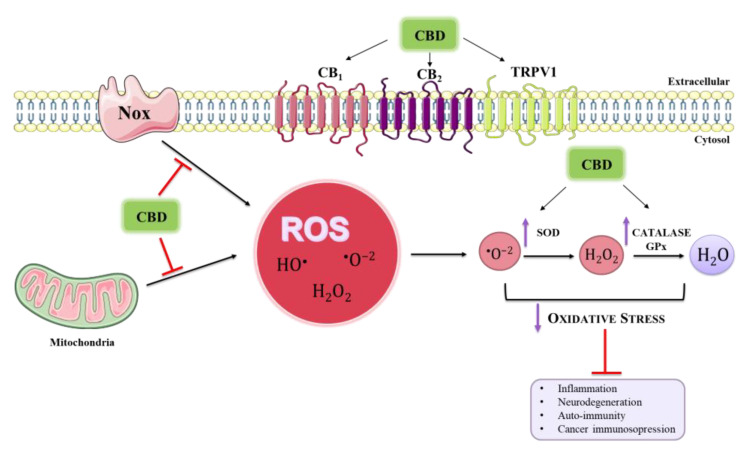
**Schematic overview of CBD inhibitory effects on ROS cellular production.** CBD modulation of oxidative stress is the basis of its effectiveness in ameliorating the symptoms of disease.

**Figure 3 ijms-24-02513-f003:**
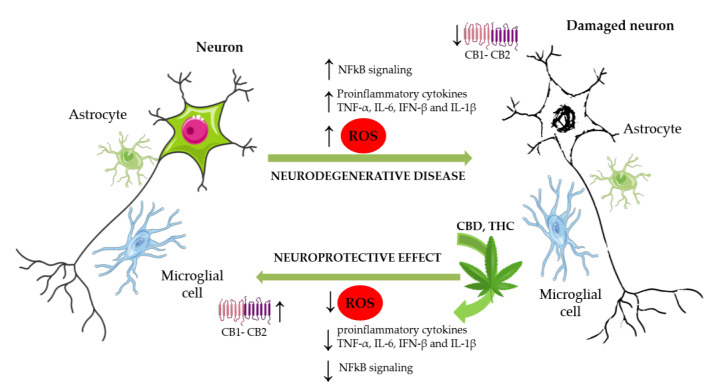
**Cannabinoids and neurodegenerative diseases.** In many neurological disorders there are incremented secretions of neurotoxic agents, such as ROS. The increment of ROS leads to NFkB activation and transduction, with the subsequent production of pro-inflammatory cytokines, such as TNF-α, IL-6, IFN-β and IL-1β. In neurological disorders, the action of CBD and THC provides neuroprotective effects through antioxidant and anti-inflammatory properties and through the activation of CB1 and CB2 to alleviate neurotoxicity.

**Table 1 ijms-24-02513-t001:** **Effects of cannabinoids in different diseases.** AD: Alzheimer’s disease; HD: Huntington’s disease; MS: Multiple sclerosis; PD: Parkinson’s disease.

CANNABINOIDS	DISEASES	EFFECTS
THC, CBD	Neurodegenerative: AD, HD, epilepsy, MS, PD	↓ ROS production [18] ↓ proinflammatory cytokines secretion [18] ↓ NFkB signaling [18] ↑ CB1-CB2 [19]
THC, CBD, AEA, 2-AG	Cancer	↓ ROS production [20] ↑ GSH [20]
CBD, CBG, ECs	Inflammatory	↓ ROS production [21] ↓ inflammation [21] ↓ oxidative/nitrosative stress [22] Alteration of XO, NOX1 and NOX4 [23]
THC, CBD, ECs	Immune system	↓ ROS production [24] ↓ neutrophil degranulation [24] ↑ CX3CR1hi macrophages [25,26] ↓ lipid peroxidation [27]

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
