# Peer review of "Cannabinoids in the Modulation of Oxidative Signaling"

_ijms, 2023, doi:10.3390/ijms24032513_

Round 1

Reviewer 1 Report

In this article the authors reviewed the effects of cannabinoids in modulation oxidative signalling. The review has it significancy in the field of potential use of cannabinoids as therapeutic approach.

Please carefully check the references:

References 1-2 are referred to the same citation. Keep attention to citation 2 in the text: does it refer to citation 1 or another one?

References from 25 to 79 are wrongly cited in the text: seems that the numbering of the reference has shifted by 1.

Here some other minor revision:

Line 76-77: Can you please rephrase for a better comprehension?

Lines 112-116: please provide a citation

Line 135: is the citation the correct one?

Lines 141-145: please provide a citation

Lines 177-179: please provide citation(s) for the multiples studies mentioned

Lines 179-181: please provide citation(s) for the experiments mentioned

Line 274: please specify the acronym IB

Line 363: please specify the acronym RA

Author Response

Dear Reviewer,

I am re-submitting the manuscript entitled: “Cannabinoids in the modulation of oxidative signaling” ijms-2150151 by Pagano et al., for consideration as a Full paper in International Journal of Molecular Sciences.

The manuscript has been revised according to the reviewer's thoughtful comments and uploaded.

Changes should be easily visible to the editors and reviewer since they were clearly highlighted.

Reviewer 1:

Point 1: References 1-2 are referred to the same citation. Keep attention to citation 2 in the text: does it refer to citation 1 or another one?

Reply: It was just a formatting error. The first citation is Pisanti, S; Bifulco, M. Medical Cannabis: plurimillennial history of an evergreen. Journal of cellular physiology vol. 234,6 (2019): 8342-8351. doi: 10.1002/jcp.27725” while the second one is “Di Marzo, V.; Bifulco, M.; De Petrocellis, L. The endocannabinoid system and its therapeutic exploitation. Nature reviews. Drug discovery. 2004, 3(9), 771–784. doi:10.1038/nrd1495” and so on. No mistake has been made in the text.

Point 2: References from 25 to 79 are wrongly cited in the text: seems that the numbering of the reference has shifted by 1

Reply: It was due to the formatting error reported in point 1. Every citation is now correct again without further modification because there was no mistake with the citations reported in the text.

Point 3: Line 76-77: Can you please rephrase for a better comprehension?

Reply: Thank you for this comment. We are glad to make modifications that will improve the fluency of our work , so we changed that line.

Point 4: Lines 112-116: please provide a citation

Reply: Thank you for this comment. We have specified the citation n° 13 “Pereira, S. R.; Hackett, B.; O'Driscoll, D. N.; Sun, M. C.; Downer, E. J. Cannabidiol modulation of oxidative stress and signalling. Neuronal signaling. 2021, 5(3), NS20200080. doi.org/10.1042/NS20200080”.

Point 5: Line 135: is the citation the correct one?

Reply: Yes, it is.

Point 6: Lines 141-145: please provide a citation

Reply: Thank you for this comment. We have specified the citation n° 17 “Dos-Santos-Pereira, M.; Guimarães, F. S.; Del-Bel, E.; Raisman-Vozari, R.; Michel, P. P. Cannabidiol prevents LPS-induced microglial inflammation by inhibiting ROS/NF-κB-dependent signaling and glucose consumption. Glia. 2020, 68(3), 561–573. doi.org/10.1002/glia.23738”

Point 7: Lines 177-179: please provide citation(s) for the multiple studies mentioned

Reply: We provide that line with a citation.

Point 8: Lines 179-181: please provide citation(s) for the experiments mentioned

Reply: We provide that line with citation.

Point 9: Line 274: please specify the acronym IB

Reply: The acronym IB is specified in line 123 and in the abbreviation list.

Point 10: Line 363: please specify the acronym RA

Reply: The acronym RA is specified in line 123 and in the abbreviation list.

Reviewer 2 Report

Cannabinoids in the modulation of oxidative signaling (IJMS)

Cannabinoids are known to act by activating specific cannabinoid receptors on the surface of cell membranes. Currently, cannabinoids are divided into three major categories: endocannabinoids, phytocannabinoids, and synthetic cannabinoids. Cannabinoids may target different mitochondrial processes (e.g., regulation of intracellular calcium levels, bioenergetic metabolism, apoptosis, and mitochondrial dynamics, including mitochondrial fission and fusion, transport, mitophagy, and biogenesis), by modulating multiple and complex signaling pathways. The main goal of this review is to investigate the potential antioxidative effects of cannabinoids compound in several diseases and conditions like neurodegenerative disorders, inflammation and pathologies of the immune system and cancer considering the increase of reactive oxygen species (ROS) reported in these pathological conditions.

This review with excellent analysis and presentation is the need of the hour. I found that the submitted review is nicely written; however, I would like to make the following suggestions:

Point 1: The authors should improve the abstract of the review article.

Point 2:   I am surprised to see that the authors have not included a section for methodology. Please add a heading for methodology in which they can provide a brief account on how they collect the data for their review.

Point 3: In the section 1.2 there is a need of add few lines about endocannabinoids and the enzymes responsible for their degradation and synthesis.

Point 4: It will be highly appreciated if the authors include a table for the review article.

Point 5: The authors should follow numbered headings and subheadings, which make the review organized and easy to understand. Correction is required; for example, the sequence is 1.2, 2.2, and 2.

Point 6: Include some critical studies that exhibit the role of Cannabinoids in oxidative stress that were not discussed in this review.

References of these studies are:

https://doi.org/10.1016/j.neuint.2020.104817; https://doi.org/10.1016/j.neuro.2011.12.015;

https://doi.org/10.1016/j.neuroscience.2014.11.016; https://doi.org/10.1016/j.phrs.2022.106603

Point 7: In section 3. Cannabinoids and ROS modulation in cancer there is a need to add recent studies that are

https://doi.org/10.3390/ijms20071673 ; 10.17305/bjbms.2018.3532; 10.1155/2018/1691428;

10.1016/j.phrs.2020.105302

Point 8: In section 2, there should be a schematic representation of the mechanism of neuroinflammation in neurodegenerative diseases.

Author Response

Dear Reviewer ,

I am re-submitting the manuscript entitled: “Cannabinoids in the modulation of oxidative signaling” ijms-2150151 by Pagano et al., for consideration as a Full paper in International Journal of Molecular Sciences.

The manuscript has been revised according to the reviewer's thoughtful comments and uploaded.

Changes should be easily visible to the editors and reviewer since they were clearly highlighted.

Reviewer 2:

Point 1: The authors should improve the abstract of the review article.

Reply: We arrange an improved abstract, aiming to highlight the key points of our work.

Point 2: I am surprised to see that the authors have not included a section for methodology. Please add a heading for methodology in which they can provide a brief account on how they collect the data for their review.

Reply: Thank you for this thoughtful comment. We selected and collected the papers based on their date of publication: we preferred recently published works over the older ones. We also selected manuscripts comprehensive of all the modern information about oxidative stress in neurodegenerative disorders, inflammation, immune system, and cancer. Surely, we are going to remember your suggestion for our future works.

Point 3: In the section 1.2 there is a need of add few lines about endocannabinoids and the enzymes responsible for their degradation and synthesis.

Reply: Thank you for this suggestion. We included some information about the enzymes responsible for their degradation and synthesis of endocannabinoids.

Point 4: It will be highly appreciated if the authors include a table for the review article.

Reply: Thank you for your exhortation. We provided the manuscript with an adjunctive table.

Point 5: The authors should follow numbered headings and subheadings, which make the review organized and easy to understand. Correction is required; for example, the sequence is 1.2, 2.2, and 2.

Reply: We corrected the numbered subheadings where necessary.

Point 6: Include some critical studies that exhibit the role of Cannabinoids in oxidative stress that were not discussed in this review.

Reply: We provide the manuscript with the new critical studies, including the ones you suggested.

Point 7: In section 3. Cannabinoids and ROS modulation in cancer there is a need to add recent studies.

Reply: We provide the manuscript with the new crucial studies, including the ones you suggested.

Point 8: In section 2, there should be a schematic representation of the mechanism of neuroinflammation in neurodegenerative diseases.

Reply: We appreciate your suggestion and we provide the manuscript with the representation requested.

Round 2

Reviewer 2 Report

I am not satisfied with the author's response to Point 2. Keeping the high-quality publication in IJMS, there is a need to add this section with the search strategies and timeline.

Author Response

Dear Reviewer, 

We apologize if we did not adequately respond to your review regarding the Methodology section. We have never found ourselves writing this section in other reviews posted on MDPi. However, we have strengthened this part of the methodology and we hope we have satisfied the request. We also think it was a great suggestion to be able to publish our review in such prestigious magazines and that we will also apply for future reviews.
Thank you.
